# Delayed Cerebral Ischemia after Aneurysmal Subarachnoid Hemorrhage: The Results of Induced Hypertension Only after the IMCVS Trial—A Prospective Cohort Study

**DOI:** 10.3390/jcm11195850

**Published:** 2022-10-02

**Authors:** Erdem Güresir, Thomas Welchowski, Tim Lampmann, Simon Brandecker, Agi Güresir, Johannes Wach, Felix Lehmann, Franziska Dorn, Markus Velten, Hartmut Vatter

**Affiliations:** 1Department of Neurosurgery, University Hospital Bonn, 53127 Bonn, Germany; 2Institute of Medical Biometry, Informatics and Epidemiology, University Hospital Bonn, 53127 Bonn, Germany; 3Department of Neuroradiology, University Hospital Bonn, 53127 Bonn, Germany; 4Department of Anesthesiology and Intensive Care, University Hospital Bonn, 53127 Bonn, Germany

**Keywords:** intracranial aneurysm, treatment, subarachnoid hemorrhage, vasospasm, delayed ischemic neurological deficit

## Abstract

Delayed cerebral ischemia (DCI) is a predictor of poor outcome after aneurysmal subarachnoid hemorrhage (SAH). Treatment strategies vary and include induced hypertension and invasive endovascular treatment. After the IMCVS trial (NCT01400360), which failed to demonstrate a benefit of endovascular treatment for cerebral vasospasm (CVS) and resulted in a significantly worse outcome, we changed our treatment policy in patients with diagnosed CVS to induced hypertension only, and we present our prospective results in the subgroup of SAH patients meeting inclusion criteria of the IMCVS trial. All patients underwent screening for DIND when conscious and for CVS using CT-A/-P at day 6–8 after SAH. In the case of CVS, arterial hypertension was induced and continued until re-assessment. In total, 149 of 303 patients developed CVS. DCI developed in 35 patients (23.5%). In multivariate analyses, CVS was a predictor for the development of new infarctions. Poor admission status, re-bleeding before treatment, and DCI predicted poor outcome. The omittance of invasive endovascular rescue therapies in SAH patients with CVS, additional to induced hypertension, does not lead to a higher rate of DCI. Potential benefits of additional endovascular treatment for CVS need to be addressed in further studies searching for a subgroup of patients who may benefit.

## 1. Introduction

Delayed cerebral ischemia (DCI) is associated with cerebral vasospasm (CVS) and is a common cause of poor outcome after aneurysmal subarachnoid hemorrhage (SAH) [1]. To improve outcome, rescue therapies including selective intraarterial infusion of vasodilators, balloon angioplasty, and induced hypertension are used [2]. Due to a lack of prospective data for the invasive diagnostic and therapeutic management of CVS, as well as the considerable inconsistency of its screening, a randomized clinical trial was initiated to assess the influences of a structured assessment of CVS and of rescue therapies on new cerebral infarctions during the phase of CVS and clinical outcome (Invasive diagnostic and therapeutic management of cerebral vasospasm after aneurysmal subarachnoid hemorrhage, IMCVS trial NCT01400360). Patients in the IMCVS trial underwent screening for CVS using MRI, performed routinely on days 4–14, and cerebral angiography between days 7–10. MRI was also performed in any case of neurological deterioration of the patient or increased mean velocity ≥150 cm/s or an increase in velocity ≥50 cm/s within 24 h in transcranial Doppler sonography. Patients with hemodynamically relevant CVS defined as 1. elevated time to peak (TTP) > 2 s compared to the corresponding contralateral side, or mean transit time (MTT) > 3.5 s, 2. profound narrowing of cerebral vessels in MRA scan, and 3. existence of “tissue at risk” (vital brain tissue with DWI lesions <50%) were randomized into a conservative group versus invasive endovascular treatment group. Conservative treatment consisted of induced hypertension (mean arterial blood pressure raised to 110 mmHg). Invasive endovascular treatment consisted of transluminal balloon angioplasty (TBA) for proximal CVS whenever possible, and/or intraarterial application of nimodipine for distal or diffuse CVS. The primary outcome measure was the development of new cerebral infarctions. After the IMCVS trial failed to demonstrate a radiological and clinical benefit of endovascular treatment, the trial was stopped prematurely because of safety concerns. In the following, we changed our treatment policy in patients with diagnosed CVS to induced hypertension only and present our prospective results in the subgroup of SAH patients meeting inclusion criteria of the IMCVS trial.

## 2. Materials and Methods

Between January 2013 and January 2020, 303 patients with SAH (WFNS 1–4), meeting the inclusion criteria of the IMCVS trial, were treated in the authors’ institution. The study was approved by the local ethics committee (Ethikkommission, University of Bonn, ID 331/12), waiving patient informed consent for the observational study.

### 2.1. Definitions and Clinical Workflow

SAH was diagnosed by computed tomography (CT) or lumbar puncture. All patients with spontaneous SAH underwent four-vessel digital subtraction angiography (DSA). Clinical data, including patient characteristics on admission and during treatment course, radiological features, and functional neurological outcome were collected and entered into a computerized database (IBM SPSS Statistics for Windows, Version 25.0. Armonk, NY, USA, IBM Corp.). Treatment decision (coil/clip) was based on an interdisciplinary approach in each individual case. Aneurysm treatment was performed within 24 h after admission. Acute hydrocephalus was treated by external cerebrospinal fluid diversion. Osmotherapy and mild hyperventilation were used for the treatment of elevated ICP (>20 mmHg). Apart from close neurological monitoring, routine surveillance included continuous invasive blood-pressure monitoring using an arterial catheter, daily transcranial Doppler measurements of red blood cell flow velocities, and ICP monitoring, as well as continuous electroencephalography in selected cases. Furthermore, CT-imaging was performed routinely (1) 24–48 h after the aneurysm clip or coil obliteration to assess procedural complications, (2) on day 14–21 to diagnose delayed cerebral infarctions and to assess the necessity of a ventriculoperitoneal shunt, and (3) at variable time points whenever neurological deteriorations occurred. All patients underwent screening for DIND when conscious, and for CVS using CT-A/CT-P at days 6–8 after SAH. In the case of CVS, arterial hypertension was induced as described in detail below and continued until re-assessment 7 days later. Sufficient fluid was administered to maintain a high normal euvolemic status. All patients received nimodipine from the day of admission. In the case of hyponatremia, fludrocortisone was added to the therapy.

### 2.2. Data recording and Clinical Outcome Measurement

Information, including patient characteristics, treatment modality, aneurysm size and location, radiological features, and neurological outcome, were assessed and further analyzed. Outcome was assessed according to the modified Rankin scale (mRS) at 6 months and stratified into favorable (mRS 0–2) versus unfavorable (mRS 3–6). For retrospective analysis of our changed therapeutic regime, patients not meeting the inclusion criteria of the IMCVS trial were excluded for further analysis.

### 2.3. Inclusion Criteria

Patient aged 18–75 years. Patients with aneurysmal SAH WFNS grades I-IV with hemodynamically relevant CVS defined as: 1. elevated time to peak (TTP) > 2 s compared to the corresponding contralateral side, or mean transit time (MTT) > 3.5 s; 2. profound narrowing of cerebral vessels of at least 50% in CT-A scan, especially in bilateral CVS; and 3. existence of “tissue at risk” (vital brain tissue with ischemic lesions <50%).

### 2.4. Induced Hypertension

Arterial hypertension, with a targeted mean arterial blood pressure (MAP) of 110 mmHg, was induced with norepinephrine and fluids via the central venous line. Induced hypertension was continued for 7 days. Thereafter, patients were re-assessed for CVS using CT-A/CT-P. In patients with persistent CVS, induced hypertension was continued for the following 7 days, and re-assessment was performed thereafter as described above. In patients with resolution of CVS, induced hypertension was terminated.

Invasive endovascular treatment (i.e., selective intraarterial infusion of vasodilators, or balloon angioplasty) was not performed at any timepoint (neither planned at a defined timepoint, nor as a rescue therapy).

### 2.5. Primary and Secondary Endpoints

The imaging data for new infarctions during the phase of CVS were assessed. The flow chart of the diagnostic protocol and the efficacy assessment are published in Vatter et al., 2011 [3].

The imaging data for new infarctions during the phase of CVS were assessed by a neuroradiologist blinded for treatment allocation. The cerebrum was partitioned into 19 segments. A 50% ischemic lesion ≥ 1 segment was defined as major, and <50% was defined as a minor infarct.

The primary outcome measure was the development of new cerebral infarctions during the phase of CVS. Secondary endpoints included clinical outcome at 6 months after SAH according to the modified Rankin scale (mRs).

### 2.6. Statistics

Data analysis was performed using the computer software package R (4.1.0). Comparisons between two groups in ordinal or continuous variables were conducted by Monte Carlo Mann–Whitney U tests. Two groups within nominal variables were compared by Monte Carlo chi-square tests. The number of repetitions in all Monte Carlo tests was set to 10^6^. The Monte Carlo variants [4] of the statistical tests were necessary because ordinal variables had many ties and contingency tables had sometimes too few cell frequencies. In those cases, the asymptotic distribution was not reliable. In multivariate analysis, the first logistic regression was used to estimate the covariate between DCI and the covariates CVS, clinical vasospasm, smoking, WFNS grade, sex, age, and re-bleeding before treatment. The second logistic regression modeled the response indicator of mRS score >3 by the covariate’s indicator of WFNS grades ≥ 3, minor and major DCI, re-bleeding before and after treatment, age, clinical vasospasm, and smoking. Before fitting the final multivariate regression models, a backward variable selection based on the AIC criterion was used.

## 3. Results

### 3.1. Patient Characteristics

We analyzed 303 patients with SAH (WFNS grades I-IV) treated in the authors’ institution; 149 patients (49.2%) developed CVS and were treated with induced hypertension.

Of the 149 patients with CVS, 35 patients experienced DCI (23.5%, Table 1). Of these, 11 patients had a concomitant delayed ischemic neurologic deficit (31.4%, DIND), and 24 patients had no evidence of DIND.

### 3.2. Aneurysm Treatment

In total, 153 patients were treated by coiling, and 150 patients by clipping.

### 3.3. Rebleeding

Six of the 303 SAH (1.9%) patients experienced early re-bleeding after aneurysm treatment. Five of the six patients with re-bleeding after treatment had been treated with coiling, and one with clipping.

### 3.4. Delayed Cerebral Ischemia

A CVS-related minor infarction was found in 18 patients, and a major infarction in 17 patients.

Nine of the 11 patients (81.8%) with CVS and concomitant DIND developed DCI. Five of the 11 patients (45.4%) with concomitant DIND developed a minor infarction, and 4 of the 11 patients (36.4%) developed a major infarction.

### 3.5. Outcome

In total, 81 of the 114 patients (71%) without DCI, compared to 13 of 35 patients (37.1%) with DCI, achieved favorable outcomes (mRs ≤ 3) after 6 months.

### 3.6. Multivariate Analyses

Of those variables analyzed for influence on the development of DCI in the univariate analyses, the variable “CVS” (*p* = 0.001; odds ratio (OR) = 26; 95% confidence interval (CI) 3.3–209) remained significant in the multivariate model (Nagelkerke’s R^2^ = 0.32). In the multivariate regression model, the variables “DIND”, “smoker”, “WFNS-grades”, “endovasular treatment”, “gender”, and “re-bleeding before treatment” were eliminated from the model.

In a second multivariate model, we analyzed variables that are known to influence clinical outcome. Of the variables that influenced poor outcome (mRs >3), the variables “WFNS grade 3–4” (*p* = 0.001; OR = 6.2; 95% CI 2.6–15), “re-bleeding before treatment” (*p* = 0.001; OR = 24.2; 95% CI 3.5–165), “DCI- minor infarction” (*p* = 0.006; OR = 4.7; 95% CI 1.6–14.6), and “DCI- major infarction” (*p* = 0.001; OR = 12.2; 95% CI 3.6–41) remained significant in the multivariate model (Nagelkerke’s R^2^ = 0.43). In the multivariate regression model, the variables “age”, “DIND”, “endovasular treatment”, “smoker”, and “re-bleeding after treatment” were eliminated from the model (Table 2).

## 4. Discussion

In the present study of SAH patients with clearly defined assessment of CVS and its therapeutic management with induced hypertension, 23.5% of SAH patients with CVS being at high risk for DCI experienced DCI, while 76.5% did not.

Despite its widespread use, there is currently no evidence that invasive endovascular therapy decreases DCI rates or improves outcome in patients with SAH. Procedure-related complications seem to outrank its benefits [5]. After the prospective IMCVS trial that failed to demonstrate a benefit from invasive endovascular rescue therapy in SAH patients with CVS, we changed our treatment policy to induced hypertension only in patients with SAH and diagnosed CVS. DCI rates vary in this vulnerable patient population suffering from CVS.

The prospective randomized trial, hypertension induction in the management of aneurysmal subarachnoid hemorrhage with secondary ischaemia (HIMALAIA) [6], did not support induced hypertension for the management of CVS. However, it was underpowered to draw definitive conclusions. Haegens et al. [7] analyzed SAH patients with diagnosed CVS retrospectively and compared DCI rates in the group of patients with induced hypertension and without. They found DCI rates of 20% in the induced hypertension group compared to 33% in the group without induced hypertension, concluding that induced hypertension seems to be effective in the prevention of DCI and consecutive poor outcome. These DCI rates of 20% in the induced hypertension group are comparable to the DCI rates of the present study (23.5%).

Jabbarli et al. [8] compared two cohorts of SAH patients with CVS/DIND treated either early or delayed with endovascular therapy and found that patients in the early treatment group experienced DCI in 20.8% versus in 29% in the group of delayed endovascular treatment. The rate of DCI in the early endovascular intervention group was similar to the DCI rates of the induced hypertension group of the present study. However, while the DCI rates are not different, any form of additional therapy may increase the rate of unintended complications. Unfortunately, complication rates are not given in all published studies. Adami et al. [5] evaluated complication rates of selective intraarterial infusion of nimodipine and percutaneous transluminal balloon angioplasty (TBA). They found new infarctions in 53% of the patients’, including 11% of patients with new infarctions that were deemed procedure-related, directly attributable to endovascular treatment (e.g., embolic complications and iatrogenic dissection). Labeyrie et al. [9] published a series of 145 patients who underwent angioplasty for CVS. They found intracranial dissections in 8%, intracranial embolism in 5%, worsening of CVS in 8%, and reperfusion syndrome with and without intracerebral hemorrhage in 5%. The study also failed to prove a clinical benefit of angioplasty in the SAH population. Additionally, to directly attributable complications, any kind of intrahospital transport for diagnostics and intervention bears the risk of transport-related complications, e.g., drop in blood pressure and lowering of blood supply below demand, resulting in ischemia. Furthermore, Hosmann et al. [10] found detrimental effects of intrahospital transports on cerebral metabolism in patients with SAH, leading to sustained impaired neuronal metabolism for several hours. The authors concluded that any kind of intrahospital transport for neuro-imaging should strongly be reconsidered and only indicated if the expected benefits outweigh the risks. It is intuitive that this also holds true for the vulnerable patient population with SAH and additional CVS.

Induced hypertension in patients with SAH and proven CVS decreases DCI rates to 20–24% (Haegens et al., and present study) [7]. The intervention is performed bedside without the necessity of repetitive intrahospital transports. Failure to respond to induced hypertension in patients with symptomatic vasospasm leads to a higher rate of cerebral infarction and poor outcome after 1 year compared to the group of patients who respond to induced hypertension [11].

In conclusion, possible benefits of any additional therapy for SAH patients with CVS treated with induced hypertension may be limited to a certain subgroup. Unfortunately, it is unclear which patient cohort this subgroup would be.

In the present study, the rate of DCI was higher in the group of patients treated endovascularly, compared to the group of patients treated surgically. While this result is surprising, it may only reflect a different distribution of known risk factors for the development of DCI, e.g., amount of cisternal and intraventricular blood, between the treatment groups. Due to the fact that treatment allocation was neither stratified according to the risk factors for DCI between the treatment groups nor randomized, no further conclusions can be drawn concerning the effect of treatment modality on DCI.

### Limitations

The most obvious limitation is the retrospective design of the study. However, after the failed IMCVS trial led by the senior author of the present study, the intention was to prospectively follow the changed treatment policy of induced hypertension only and its results. Further benefits of the study are the clear definition and screening of CVS using CT-A/CT-P at days 6–8 after SAH. By analyzing rates of DCI, however, it is clear that by including high- and low-grade SAH patients the assessments of neurological status in awake and sedated patients may have been different.

## 5. Conclusions

In conclusion, we found that the omittance of invasive endovascular rescue therapies, additional to induced hypertension, in SAH patients with CVS does not lead to a higher rate of delayed cerebral ischemia compared to data from the literature and to data of the conservative treatment group of the IMCVS trial. Potential benefits of additional endovascular treatment for CVS need to be addressed in further studies, searching for a subgroup of patients who may benefit.

## Figures and Tables

**Table 1 jcm-11-05850-t001:** Patient characteristics.

Variable	DCI (n = 35)	No DCI (n = 114)	*p*-Value
Age, Y ± SD, mean	54 ± 11	55 ± 14	0.2
Female gender (%)	22 (63)	66 (58)	0.6
Smoker (%)	14 (40)	52 (46)	0.9
mRS before SAH	0	0	0.9
Hydrocephalus at admission (%)	27 (77.1)	73 (64)	1.0
WFNS grade	2 ±1	2 ± 1	0.9
Fisher score	3	3	0.9
IVH	8	34	1.0
ICH < 3 cm	2	15	0.1
ICH > 3 cm	0	4	0.1
Aneurysm size	6 ± 3	7 ± 4	0.051
Warning leak	1	5	0.4
Coiling/Clipping	23/12	49/65	0.02
mRs ≤ 3 after 6 months	13 (37.1%)	81 (71%)	<0.001

ICH = intracerebral hemorrhage; IVH = intraventricular hemorrhage; mRS = modified Rankin scale; SAH = subarachnoid hemorrhage; Y = years.

**Table 2 jcm-11-05850-t002:** Multivariate analyses.

Variable	OR	CI	*p*-Value
Multivariate analysis: Predictors of DCI (Nagelkerke’s R^2^ = 0.32)			
CVS	26	3.3–209	**0.001**
DIND	2.5	0.06–5.3	0.6
Smoker	4	0.3–64	0.3
WFNS-grades	2.4	0.4–18	0.4
Endovascular treatment	1.3	0.6–2.6	0.4
Gender	2.5	0.4–15	0.3
Re-bleeding before treatment	2.5	0.06–104	0.6
Multivariate analysis: Predictors of poor outcome (Nagelkerke’s R^2^ = 0.43)			
WFNS grade 3–4	6.2	2.6–15	**0.001**
Re-bleeding before treatment	24.2	3.5–165	**0.001**
DCI-minor infarction	4.7	1.6–14.6	**0.006**
DCI-major infarction	12.2	3.6–41	**0.001**
Age	0.96	0.93–1.1	0.09
DIND	2	0.9–4.6	0.1
Endovascular treatment	0.5	0.08–3.7	0.5
Smoker	1.2	0.5–3	0.7
Re-bleeding after treatment	4.8	0.6–38	0.1

CI = 95% confidence interval; CVS = cerebral vasospasm; DCI = delayed cerebral ischemia; DIND = delayed ischemic neurologic deficit; OR = odds ratio; WFNS = World Federation of Neurosurgical Societies; *p*-values < 0.05 are marked bold.

## Data Availability

All data are included in this manuscript.

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
