# Peer review of "Delayed Cerebral Ischemia after Aneurysmal Subarachnoid Hemorrhage: The Results of Induced Hypertension Only after the IMCVS Trial—A Prospective Cohort Study"

_jcm, 2022, doi:10.3390/jcm11195850_

Round 1

Reviewer 1 Report

Authors conducted a retrospective review of a cohort of patients who were treated with high blood pressure following the diagnosis of CVS. I agree with the authors that there is no evidence that invasive endovascular treatment is beneficial for such patients. Regardless not obtaining favourable results in the trials, it seems that arterial hypertension could be useful in selected cases. The article is well written and presents a correct methodology. I have some concerns that perhaps the authors can answer me:

1/ It is surprising that aneurysm exclusion treatment (coiling/clipping) could have an effect on the onset of ICD. It seems to have a protective effect, although it should be fallen in multivariate analysis? Perhaps they could refer to this topic in the discussion where it does not seem to stand out despite a remarkable result.

2/ The description of multivariate analysis is somewhat confusing. I understand that two models are finally created, although I do not fully understand which variables remain in these models. Perhaps a table with the two models (with CI, R2 and variables could help identify variables that could be useful).

Author Response

Reviewer 1:

1/ It is surprising that aneurysm exclusion treatment (coiling/clipping) could have an effect on the onset of ICD. It seems to have a protective effect, although it should be fallen in multivariate analysis? Perhaps they could refer to this topic in the discussion where it does not seem to stand out despite a remarkable result.

We included the following into the discussion (p.7)

In the present study, the rate of DCI was higher in the group of patients treated endovascularly, compared to the group of patients treated surgically. While this result is surprising, it may only reflect a different distribution of known risk factors for the development of DCI, e.g. amount of cisternal and intraventricular blood, between the treatment groups. Due to the fact that treatment allocation was neither stratified according to the risk factors for DCI between the treatment groups, nor randomized, no further conclusions can be drawn concerning the effect of treatment modality on DCI.

2/ The description of multivariate analysis is somewhat confusing. I understand that two models are finally created, although I do not fully understand which variables remain in these models. Perhaps a table with the two models (with CI, R2 and variables could help identify variables that could be useful).

We tried to clarify and included “Table 2” into the section “3.6 Multivariate analyses”:

          TABLE 2. Multivariate analyses

Variable

OR

CI

p-value

Multivariate analysis: Predictors of DCI (Nagelkerke’s R2 = 0.32)

CVS

26

3.3 – 209

0.001

DIND

2.5

0.06 – 5.3

0.6

Smoker

4

0.3 – 64

0.3

WFNS-grades

2.4

0.4 – 18

0.4

Endovascular treatment

1.3

0.6 – 2.6

0.4

Gender

2.5

0.4 – 15

0.3

Re-bleeding before treatment

2.5

0.06 – 104

0.6

Multivariate analysis: Predictors of poor outcome (Nagelkerke’s R2 = 0.43)

WFNS grade 3-4

6.2

2.6 – 15

0.001

Re-bleeding before treatment

24.2

3.5 – 165

0.001

DCI- minor infarction

4.7

1.6 – 14.6

0.006

DCI- major infarction

12.2

3.6 – 41

0.001

Age

0.96

0.93 – 1.1

0.09

DIND

2

0.9 – 4.6

0.1

Endovascular treatment

0.5

0.08 – 3.7

0.5

Smoker

1.2

0.5 – 3

0.7

Re-bleeding after treatment

4.8

0.6 – 38

0.1

CI = 95% Confidence Interval; CVS = Cerebral Vasospasm; DCI = Delayed Cerebral Ischemia; DIND = Delayed Ischemic Neurologic Deficit; OR = Odds Ratio; WFNS = World Federation of Neurosurgical Societies; p-values <0.05 are marked bold

Reviewer 2 Report

Authors report the results of induced hypertension in patients with SAH and significant cerebral vasospasm. This treatment strategy has been selected by the authors after the results of a previous unpublished study of SAH patients with cerebral vasospasm who were randomized to receive medical therapy vs endovascular therapy (using different endovascular approaches, being angioplasthy one of them). Making a short summary of the IMCVS trial in the introduction or methods section will be of use to readers.  

The study design is a retrospective review of patients admitted to a single institution. One of the drawbacks of the design is the inclusion of low grade and high grade SAH patients. Neurological status in intubated patients cannot be assessed in the same way as in awake patients. I believe this point should be raised in the limitations section and authors should explain if they have used multimodality monitoring in sedated patients. 

Moreover, some aspects of the study methods need to be clarified: 

- Please, define CVS: could authors state the degree of vessel stenosis they have used to define significant CVS?

- Please, explain how has blood pressure been monitored: continuous invasive blood pressure monitoring using an arterial catheter or non-invasive blood pressure monitoring. 

- How many patients were intubated and did not permit a close neurological monitoring?

- In patients with no response to induced hypertension, was any rescue therapy considered?

Author Response

Reviewer 2:

Making a short summary of the IMCVS trial in the introduction or methods section will be of use to readers.

To introduce the readers to the IMCVS trail we revised section “1. Introduction” and added:

“Due to a lack of prospective data for invasive diagnostic and therapeutic management of CVS, as well as considerable inconsistency of its’ screening, a randomized clinical trial was initiated in order to assess the influence of a structured assessment of CVS, and of rescue therapies on new cerebral infarctions during the phase of CVS and clinical outcome (Invasive diagnostic and therapeutic Management of Cerebral VasoSpasm after aneurysmal subarachnoid hemorrhage, IMCVS trial NCT01400360). Patients in the IMCVS trial underwent screening for CVS using MRI, performed routinely on day 4-14, and cerebral angiography between day 7-10. MRI was also performed in any case of neurological deterioration of the patient or increased mean velocity ³ 150 cm/s or an increase in velocity ³ 50cm/s within 24 h in transcranial Doppler sonography. Patients with hemodynamically relevant CVS defined as 1. Elevated time to peak (TTP) > 2 seconds compared to the corresponding contralateral side, or mean transit time (MTT) > 3.5 seconds. 2. Profound narrowing of cerebral vessels in MRA scan. 3. Existence of “tissue at risk” (vital brain tissue with DWI lesions <50%) were randomized into a conservative group versus invasive endovascular treatment group. Conservative treatment consisted of induced hypertension (mean arterial blood pressure raised to 110 mmHg). Invasive endovascular treatment consisted of transluminal balloon angioplasty (TBA) for proximal CVS whenever possible, and / or intraarterial application of nimodipine for distal or diffuse CVS. Primary outcome measure was the development of new cerebral infarctions.

“The IMCVS trial aimed to compare outcome after transluminal Balloon angioplasty (TBA) or intraarterial application of vasodilators when CVS occurred to conventional treatment without intraarterial therapy.”

The study design is a retrospective review of patients admitted to a single institution. One of the drawbacks of the design is the inclusion of low grade and high grade SAH patients. Neurological status in intubated patients cannot be assessed in the same way as in awake patients. I believe this point should be raised in the limitations section and authors should explain if they have used multimodality monitoring in sedated patients.

The reviewer is absolutely right that the neurological status of patients under sedation cannot be assessed equally to awake patients. For that reason, all patients underwent screening for DIND when conscious, and for CVS using CT-A / CT-P at day 6-8 after SAH, as stated in the methods section.

We included the following into the section “4.1 Limitations” to specify that: Further benefits of the study are the clear definition and screening of CVS using CT-A / CT-P at day 6-8 after SAH. By analyzing rates of DCI However, it is clear that including high and low grade SAH patients, the assessment of neurological status in awake and sedated patients, may have been different.

Furthermore we included the following into the methods section (p.2):

“Apart from close neurological monitoring routine surveillance included continuous invasive blood pressure monitoring using an arterial catheter, daily transcranial Doppler measurements of red blood cell flow velocities, ICP monitoring, as well as continuous electroencephalography in selected cases. Furthermore, CT-imaging was performed routinely 1) 24-48 hours after aneurysm clip or coil obliteration to assess procedural complications, 2) on day 14-21 to diagnose delayed cerebral infarctions and to assess the necessity of a ventriculoperitoneal shunt, and 3) at variable time points whenever neurological deteriorations occurred.”

Moreover, some aspects of the study methods need to be clarified:

- Please, define CVS: could authors state the degree of vessel stenosis they have used to define significant CVS?

The majority of patients with clinically relevant CVS had elevated time to peak (TTP) > 2 seconds compared to the corresponding contralateral side, or mean transit time (MTT) > 3.5 seconds as stated in section “2.3 Inclusion criteria”. In patients with bilateral CVS, narrowing of at least 50% of cerebral arteries, attributable to CVS, was used to define relevant CVS.

We revised this section: “2. Profound narrowing of cerebral vessels of at least 50 % in CT-A scan, especially in bilateral CVS.”

- Please, explain how has blood pressure been monitored: continuous invasive blood pressure monitoring using an arterial catheter or non-invasive blood pressure monitoring.

Continuous invasive blood pressure monitoring with an arterial catheter was used to monitor blood pressure in all patients. We revised section “2.1 Definitions and clinical workflow” to state this: “Apart from close neurological monitoring routine surveillance included continuous invasive blood pressure monitoring using an arterial catheter, daily transcranial Doppler measurements of red blood cell flow velocities, ICP monitoring, as well as continuous electroencephalography in selected cases. Furthermore, CT-imaging was performed routinely 1) 24-48 hours after aneurysm clip or coil obliteration to assess procedural complications, 2) on day 14-21 to diagnose delayed cerebral infarctions and to assess the necessity of a ventriculoperitoneal shunt, and 3) at variable time points whenever neurological deteriorations occurred.”

- How many patients were intubated and did not permit a close neurological monitoring?

Unfortunately, we can’t answer this question. Aneurysm treatment is always performed in intubated patients, and patients were extubated as soon as possible. However, especially poor grades, some of the patients remain or get intubated until the start clinically relevant CVS / DCI. That’s exactly the reason, why we screen for CVS in a quite aggressive manner (clinically if possible, TCD, routine CT-A / CT-P at day 6-8). Further imaging studies were performed whenever deemed necessary.

- In patients with no response to induced hypertension, was any rescue therapy considered?

As the IMCVS trial failed to demonstrate a radiological benefit of endovascular treatment and may even point to poorer clinical outcome in the whole cohort of SAH patients who underwent invasive endovascular treatment, we did not apply any rescue therapy.

We revised section “2.4 Induced hypertension” to clarify this: “Invasive endovascular treatment (i.e., selective intraarterial infusion of vasodilators, or balloon angioplasty) was not performed at any timepoint (neither planned at a defined timepoint, nor as a recue therapy).”

Reviewer 3 Report

This is a quite intriguing study on the use of induced hypertension for cerebral vasospasm (CVS) after SAH in 149 patients. Following the negative results of the IMCVS trial (invasive/endovascular management of CVS), the authors changed their treatment algorithm, and they provide a detailed retrospective review of their results with induced hypertension only.

One of the strengths of the study is their itemized definition of DCI, DIND and CVS, which is always a controversial topic on this field. The usually ambiguous definition of all these entities varies widely from one article to another, making it difficult to compare results between studies. Their detailed definition provides accurate data easily applicable to most clinical settings.

The article is well written, the methodology is widely detailed, the discussion adds further value to current knowledge and the conclusions tally the authors’ results.

As a minor concern, this reviewer wonders about the apparent effect of clipping as a protective factor for DCI on the univariate analysis (according to table 1). I guess it disappears on the multivariate analysis, but it might be interesting to provide a short discussion/explanation on this.  

Author Response

Reviewer 3:

This is a quite intriguing study on the use of induced hypertension for cerebral vasospasm (CVS) after SAH in 149 patients. Following the negative results of the IMCVS trial (invasive/endovascular management of CVS), the authors changed their treatment algorithm, and they provide a detailed retrospective review of their results with induced hypertension only.

One of the strengths of the study is their itemized definition of DCI, DIND and CVS, which is always a controversial topic on this field. The usually ambiguous definition of all these entities varies widely from one article to another, making it difficult to compare results between studies. Their detailed definition provides accurate data easily applicable to most clinical settings.

The article is well written, the methodology is widely detailed, the discussion adds further value to current knowledge and the conclusions tally the authors’ results.

As a minor concern, this reviewer wonders about the apparent effect of clipping as a protective factor for DCI on the univariate analysis (according to table 1). I guess it disappears on the multivariate analysis, but it might be interesting to provide a short discussion/explanation on this.

We would like to thank the reviewer for his thoughtful comments.

In order to clarify, we discussed the topic of DCI and treatment modality at p.7:

In the present study, the rate of DCI was higher in the group of patients treated endovascularly, compared to the group of patients treated surgically. While this result is surprising, it may only reflect a different distribution of known risk factors for the development of DCI, e.g. amount of cisternal and intraventricular blood, between the treatment groups. Due to the fact that treatment allocation was neither stratified according to the risk factors for DCI between the treatment groups, nor randomized, no further conclusions can be drawn concerning the effect of treatment modality on DCI.

Furthermore, we included a new table (Table 2) with the results of the multivariate analyses. Treatment modality (“Endovascular treatment”) is no predictor of DCI, and no predictor of poor outcome in the present study.

We would like to thank the reviewers for thoroughly reviewing our manuscript and their helpful comments. We believe that after incorporating all the issues listed above, the manuscript is clearer and the study is now better understood. We hope that the manuscript now is eligible for publication.

Round 2

Reviewer 2 Report

Authors have addressed all points raised in the previous review.